# MRI in Pregnancy and Precision Medicine: A Review from Literature

**DOI:** 10.3390/jpm12010009

**Published:** 2021-12-23

**Authors:** Gianluca Gatta, Graziella Di Grezia, Vincenzo Cuccurullo, Celestino Sardu, Francesco Iovino, Rosita Comune, Angelo Ruggiero, Marilena Chirico, Daniele La Forgia, Annarita Fanizzi, Raffaella Massafra, Maria Paola Belfiore, Giuseppe Falco, Alfonso Reginelli, Luca Brunese, Roberto Grassi, Salvatore Cappabianca, Luigi Viola

**Affiliations:** 1Section of Radiology and Radiotherapy, Department of Precision Medicine, University of Campania Luigi Vanvitelli, 80138 Naples, Italy; ggatta@sirm.org (G.G.); rositacomune1993@libero.it (R.C.); mariapaolabelfiore@gmail.com (M.P.B.); alfonso.reginelli@unicampania.it (A.R.); roberto.grassi@unicampania.it (R.G.); salvatore.cappabianca@unicampania.it (S.C.); 2Radiology Department, G. Criscuoli Hospital, 83054 Sant’Angelo dei Lombardi, Italy; graziella.digrezia@unicampania.it; 3Nuclear Medicine Unit, Department of Precision Medicine, University of Campania Luigi Vanvitelli, 80138 Naples, Italy; vincenzo.cuccurullo@unicampania.it; 4Department of Advanced Medical and Surgical Sciences, University of Campania Luigi Vanvitelli, 80138 Naples, Italy; drsarducele@gmail.com; 5Department of Translational Medical Science, School of Medicine, University of Campania Luigi Vanvitelli, 80138 Naples, Italy; francesco.iovino@unicampania.it; 6Department of Public Health, Pharmaceutical Use and Dermatology, University of Naples Federico II, 80138 Naples, Italy; angeloruggiero1993@libero.it; 7Department of Anesthesiology and Intensive Care Medicine, University of Naples Federico II, 80138 Naples, Italy; chirico.marilena@gmail.com; 8Department of Breast Radiology, Giovanni Paolo II/I.R.C.C.S. Cancer Institute, 70124 Bari, Italy; d.laforgia@oncologico.bari.it (D.L.F.); annarita.fanizzi.af@gmail.com (A.F.); r.massafra@tiscali.it (R.M.); 9Italian Society of Medical and Interventional Radiology (SIRM), SIRM Foundation, 20122 Milan, Italy; 10Breast Surgery Unit, Department of Oncology and Advanced Technologies, IRCSS Santa Maria Nuova Hospital, 42123 Reggio Emilia, Italy; giuseppe.falco@ausl.re.it; 11Medicine and Health Science Department, University of Molise, 86100 Campobasso, Italy; luca.brunese@unimol.it

**Keywords:** pregnancy, MRI, gadolinium, liposomal gadolinium, safety during pregnancy, precision medicine

## Abstract

Magnetic resonance imaging (MRI) offers excellent spatial and contrast resolution for evaluating a wide variety of pathologies, without exposing patients to ionizing radiations. Additionally, MRI offers reproducible diagnostic imaging results that are not operator-dependent, a major advantage over ultrasound. MRI is commonly used in pregnant women to evaluate, most frequently, acute abdominal and pelvic pain or placental abnormalities, as well as neurological or fetal abnormalities, infections, or neoplasms. However, to date, our knowledge about MRI safety during pregnancy, especially about the administration of gadolinium-based contrast agents, which are able to cross the placental barrier, is still limited, raising concerns about possible negative effects on both the mother and the health of the fetus. Contrast agents that are unable to cross the placenta in a way that is safe for the fetus are desirable. In recent years, some preclinical studies, carried out in rodent models, have evaluated the role of long circulating liposomal nanoparticle-based blood-pool gadolinium contrast agents that do not penetrate the placental barrier due to their size and therefore do not expose the fetus to the contrast agent during pregnancy, preserving it from any hypothetical risks. Hence, we performed a literature review focusing on contrast and non-contrast MRI use during pregnancy.

## 1. Introduction

In recent years, due to a greater availability of imaging resources, technological advances in diagnostic imaging, and an increase in forensic litigation, there has been an exponential increase for medical imaging.

However, this type of examination requires a series of profound reflections as they involve the health of both the pregnant woman and the fetus, raising a series of medical, ethical, and legal assessments [1,2]. Ultrasound (US) and magnetic resonance imaging (MRI) are the most commonly used imaging modalities in pregnancy as they lack ionizing radiation. Compared to US, MRI also has the advantages of not being operator-dependent, and of providing greater anatomical details, due to the continuous progress made since its advent in the mid-1980s [3,4]. Particularly, the use of MRI, as also suggested by the guidelines proposed by the American Congress of Obstetricians and Gynaecologists (ACOG), is recommended when the ultrasound examination shows unclear results as it can improve diagnostic accuracy, especially in cases of posterior localization of the placenta or abnormally invasive placenta (AIP), by visualizing the utero-placental interface [5,6]. Moreover, during pregnancy there are various physiological changes that may have an influence on MRI. Indeed, during pregnancy, the abdomen is the site of profound anatomical changes. The uterus increases its size with the passing of the weeks of gestation, becoming an abdominal organ as early as the second trimester [7]. As a result, all abdominal organs undergo compression from the uterus [8]. In particular, the hollow organs are the ones that are most affected by compression from the uterus: the stomach is pushed more cranially, the intestine laterally, and the bladder more caudally [7,8]. The diaphragm also suffers from the reduction of space at the abdominal level, being pushed cranially by at least 4 cm. The veins, whose wall is more compressible than that of the arteries, are also affected by the increase in size of the uterus [7]. In particular, a flattening of the lower third of the vena cava is often observed in the last weeks of gestation. All these aspects should be considered during MRI evaluation in pregnancy. Herein, we performed a review focusing on contrast and non-contrast MRI use during pregnancy.

## 2. Methods

Literature review of the scientific literature of case clinical and preclinical studies, and case reports, regarding the use of MRI with and without contrast during pregnancy was performed.

PubMed was searched using terms “MRI AND pregnancy”, “Gadolinium contrast agents AND pregnancy”, “MRI AND fetus”, “MRI safety during pregnancy”, “MRI effects on fetus”, “MRI”, “Liposomal gadolinium”, and “precision medicine in pregnancy”. Search criteria were the following: (a) articles published in scientific journals included in MEDLINE or EMBASE databases; (b) articles written in English. All types of epidemiological studies were included; narrative reviews were excluded. Title and abstract review were performed. Case reports and case series were included in this review. The article is based on previously conducted studies.

## 3. Body

### 3.1. Non Contrast MRI during Pregnancy

To date, few data have been reported about eventual effects of MRI during pregnancy. There are theoretical risks regarding the process of deposition of energy in the body in the form of heat, which is quantified by the specific absorption ratio (SAR), measured in units of watts per kilogram (W/kg). In animal models, it has been observed that tissue heating caused by elevated SAR during pregnancy resulting in an increase in maternal body temperature of more than 2–2.5 °C for at least 30–60 min causes fetal harm [9]. In light of this, the Food and Drug Administration (FDA) advises, in clinical practice, not to exceed the maximum SAR for the whole body of 4 W/kg, which is capable of increasing the body temperature by 0.6 °C for 30 min of MRI. It was also observed that the heating of the tissues is lower in the deep tissues, where the fetus is located, compared to the maternal body surface. Therefore, observing the SAR limits imposed, the heating of the tissues is not considered a serious risk factor for the fetus. Therefore, the International Electrotechnical Commission (IEC), as a precaution, to reduce the effects of tissue heating, imposes a limit for pregnant patients of whole-body SAR of 2 W/kg [10].

During the first trimester of pregnancy, fetal cells proliferate, differentiate, migrate and implant, going through one of the most crucial phases of pregnancy, namely, organogenesis.

Precisely during this delicate phase, the main risks are related to an altered organogenesis or to a possible miscarriage [11].

Although several in vitro studies on mammals stem cells have shown that exposure to MRI influences cell proliferation, differentiation, and migration, via altered cell signalling [12,13], and that in animal models during pregnancy was associated with reduced birth weight and increased stillbirth [14], to date, no observational studies in humans have shown adverse effects, such as teratogenic effects, or differences in birth weight or perinatal mortality rate, of MRI on the fetus during pregnancy (as well as on children born to pregnant women exposed to MRI). However, major limitations of available human studies are their retrospective nature and the lack of long-term data [15,16,17,18,19,20].

According to the American College of Radiology (ACR) and ACOG guidelines, MRI, performed with 3.0 T scanners or less, is not associated with any adverse effects on the fetus, but it should be used prudently in any gestational ages [21,22]. Hence, MRI is recommended if the information provided may affect the medical treatment of the pregnant woman or fetus, if it is not possible to wait for the term of pregnancy, and if it is not possible to perform an alternative method that does not use ionizing radiation, such as US. Furthermore, the exposure, compatibly with the pursuit of the pre-established diagnostic goals, must be as short as possible [23].

### 3.2. Gadolinium-Based Contrast Agents (GBCAs) MRI during Pregnancy

Gadolinium-based contrast agents (GBCAs) are intravenously administered contrasts for MRI approved for clinical use in patients over 20 years. These agents enhance the clarity and detection of images, improving diagnoses [24]. Chemically, there are currently two different types of GBCAs: linear contrast agents and macrocylic contrast agents (Table 1). Macrocylic contrast agents appear to have lower dissociation constants and lower retention within the body than linear agents [24,25]. Gadolinium, used in about one third of MRI exams, is toxic in its free ionic form (gadolinium 3+) but biologically inert in its complexed form, which is why chelates to a ligand (GBCA) are used [26,27,28].

### 3.3. Risks Related to GBCA Administration

The literature suggests that both short and long-term risks after GBCA administration were observed in pregnant patients as in the general population; however, reactions to contrast agents that are unique to pregnancy have also been reported [29]. Short-term risks include allergic reactions and non-allergic reactions, such as nausea and vomiting. However, there are severe reactions to the contrast agent that are characteristic of pregnancy, such as recurrent late decelerations, prolonged fetal bradycardia on fetal heart tracing, and preterm labor [30,31,32].

Long-term risks include nephrogenic systemic fibrosis (NSF) and retained intracranial gadolinium.

NSF is a rare and debilitating disease characterized by fibrosing skin lesions and organ failure, observed in patients with impaired renal function. It was first described in 2000, but only in 2006 was it related to the intravenous administration of GBCA. To date, however, no cases of NSF have been reported in a pregnant patient or newborn after intrauterine exposure [33,34]. The retained intracranial gadolinium, first described as observed T1 shortening predominantly in the globus pallidus and dentate nucleus, and also observed in patients with normal renal function, has been related to multiple administrations of GBCA during the life, leading to greater caution in the use of the contrast agent [35,36]. Moreover, subsequent biopsy and autopsy-based studies revealed retained gadolinium in other parts of the body, including the bones, the skin, the liver, and the bone marrow, following the use of mainly linear but also macrocyclic agents, in a dose-dependent manner. To date, however, no symptoms have been observed following retained gadolinium, whose clinical significance remains uncertain [37,38,39]. Although human studies performed during pregnancy are still lacking, the deposition of gadolinium in the fetus is of particular interest due to the rapid development of the brain and other organs during this period, as well as a greater probability of undergoing further administration during the course of life [40,41]. Notably, recent studies examining the degree of gadolinium deposition associated with in-utero exposure in mammalian animal models have shown detectable concentrations of gadolinium in the brain, bone, and liver [42,43]. Hence, intravenous administration of clinically approved GBCAs, although not contraindicated during pregnancy, should be avoided unless necessary, such as when the potential benefits outweigh the risks. Its use should therefore be assessed on a case-by-case basis.

### 3.4. Pharmacokinetic Studies of GBCAs in Animal Models

Pharmacokinetic studies, evaluating trans-placental passage of gadolinium in animal models, demonstrated the ability of GBCAs to cross the placenta [44,45]. After maternal intravenous injection, chelated gadolinium was in fact detected in amniotic fluid and fetal tissues after 24–48 h [46,47]. Although GBCA administration is known to expose the fetus to gadolinium chelate in animals, the possible teratogenic effects of GBCAs on the fetus are still unclear. Indeed, the literature reported contrasting results derived from studies carried out on animal models. Particularly, some studies showed a higher rate of spontaneous abortion, reduced mean birth weight, and congenital anomalies after administration of GBCA at high and repeated doses over time (supra-clinical/supra-therapeutic), while others showed no harmful effects [48,49].

Gadolinium-induced toxicity arises primarily from the dissociation of free gadolinium from the chelated one. Normally, the plasma half-life of GBCAs in maternal blood, in patients with normal renal function, is about 2 h, with total excretion over 24 h. The chelated gadolinium crosses the placenta and enters the fetal blood circulation [46,47]; the GBCA is then filtered by the fetal kidneys and excreted in the amniotic fluid, and finally it passes partly into the maternal blood circulation and partly into the digestive system of the fetus after swallowing amniotic fluid, from where it enters in the fetal circulation again [48,50].

### 3.5. GBCAs in Humans

As regards GBCAs use in humans during pregnancy, a retrospective study on a limited sample of patients evaluated the effect of administering GBCA during the first trimester of pregnancy in 26 patients, showing no adverse perinatal or neonatal effects [51]. In contrast, a larger 4-year-long prospective study, conducted in the Canadian province of Ontario, including 397 pregnancies undergoing an MRI examination with GBCA administration, and compared with over 1.4 million controls, found that fetal exposure to gadolinium, particularly during the second or third trimester of pregnancy, was associated with a higher incidence rate of stillbirth (approximately 1%), and of cutaneous and rheumatic diseases, such as dermatitis, vasculitis, and arthritis, in the exposed group compared to the unexposed group [20]. However, further studies are needed to replicate these findings and address the different limitations of the study [20]. On the basis of these results, the FDA had rated GBCAs as Category C, “Animal reproduction studies have shown an adverse effect on the fetus and there are no adequate and well-controlled studies in human beings, but potential benefits may justify the use of the drug in pregnant women despite the potential risks” since a teratogenic effect of contrast agents was found, in vivo, in animal models when administered at high doses [49,52], but at the same time there are no in-depth and clear studies on humans [53,54,55]. With regard to the official guidelines of scientific societies about the use of contrast MRI during pregnancy, the ACR, the European Society of Urogenital Radiology (ESUR), and the Royal College of Radiology (RCR), in light of the fact that GBCAs are able to cross the placenta and to date there is no clear evidence on safety for the fetus, suggest an even more cautious approach in the use of GBCAs during pregnancy [56,57,58]. Hence, GBCAs should only be used if the potential benefits for the pregnant woman or the fetus outweigh the potential risks. Before any MRI, an accurate analysis of the risk-benefit ratio by the radiologist and the referring physician is required. It is therefore necessary to clearly explain to the patient any risks and benefits [56,57,58].

### 3.6. Main Indications for Emergency MRI during Pregnancy

The most common clinical indications for an emergency MRI during pregnancy include both maternal and fetal conditions. In all these cases, US usually represents the first diagnostic choice; however, if the US does not lead to a diagnosis of certainty, it may be useful to resort to MRI [29,30].

### 3.7. Main Maternal Indications for Emergency MRI during Pregnancy

The main maternal indications MRI during pregnancy may be divided into obstetric and non-obstetric causes.

### 3.8. Obstetric Causes

#### 3.8.1. Abnormally Invasive Placenta

Invasive placentation occurs in 1 out of 2500 pregnancies and may lead to significant bleeding after delivery with high maternal mortality rate. The main risk factors are represented by previous placenta previa, advanced maternal age, multiparity, previous caesarean section, previous uterine curettage, previous cycles of uterine radiotherapy, presence of uterine leiomyoma, uterine malformations, hypertensive disorders of pregnancy, and smoking habit. Although it is usually diagnosed initially by the US, MRI appears to offer a better visualization of areas of abnormal placentation [59,60].

#### 3.8.2. Placental Abruption

Placental abruption, classically defined as a premature separation of the placenta before delivery, represents a major cause of vaginal bleeding in the second half of pregnancy and is characterized by elevated maternal, fetal, and neonatal morbidity and mortality rates. In developed countries, placental abruption occurs in approximately 1% of pregnancies and is the cause of approximately 10–20% of all perinatal deaths. Symptoms are typically characterized by intense abdominal pain and severe vaginal bleeding. An early diagnosis is crucial in its management. US represents the first diagnostic method choice; however, in doubtful cases it is useful to resort to MRI to better identify the hematomas, characterize the age, and differentiate hematomas from tumors. Hematomas are classified based on the location as retroplacental, marginal subchorionic, preplacental (subamniotic), or intraplacental [61,62].

#### 3.8.3. Uterine Rupture

Uterine rupture results in a complete separation of the uterine serosa and myometrium. It is a very rare event (1 in 5000–20,000 pregnancies), usually occurring after a dehiscence of a caesarean section scar or trauma. It is linked to high mortality rate for both the mother and the fetus. The symptomatology is characterized by violent and intense pain [63,64,65].

#### 3.8.4. Ovarian Cysts/Ovarian Torsion

About 2% of women during pregnancy have ovarian cysts. Most of these are simple asymptomatic cysts, often less than 5 cm in size, which resolve spontaneously. A small percentage of them (1–3%), however, may have a malignant nature. When adnexal cysts are larger than 4 cm, there is a greater risk of ovarian torsion (about 1 pregnancy in 1800), which occurs more frequently at early pregnancy. Symptoms are characterized by acute, sometimes intermittent, pelvic pain [66,67].

#### 3.8.5. Other Obstetric Causes

Other obstetric causes of abdominal pain during pregnancy, which may require MRI, include ectopic pregnancy, degenerating leiomyoma, gonadal vein dilatation, and neoplasia [68,69].

### 3.9. Non Obstetric Causes

#### 3.9.1. Acute Abdominal Pain

There are several causes related to the onset of acute abdominal pain during pregnancy, both obstetric and non-obstetric. The physiological and marked anatomical variations, combined with a physical examination of the abdomen hindered by the presence of the fetus and a physiological leukocytosis during this period, can lead to a diagnostic delay, with severe consequences for both the mother and the fetus. If the US does not lead to a diagnosis of certainty, it is useful to resort to MRI for the greater diagnostic accuracy of the fetus, placenta, uterus, and abdominal organs [67,70,71].

#### 3.9.2. Acute Appendicitis

Acute appendicitis is one of the most common causes of an acute abdomen in pregnancy, occurring in approximately 1 case out of 1500 pregnancies, with an overall incidence of 0.05% to 0.07%, or the same incidence observed in the non-pregnant population. The most common presenting symptoms are represented by nausea, vomiting, anorexia, right lower quadrant pain, and uterine contractions, and less frequently during pregnancy, fever, and tachycardia. Among the diagnostic imaging techniques, US has been shown to be very sensitive when performed by experienced sonographers (ranging from 12.5 to 100%). However, the physiological upward displacement of the appendix during the last weeks of pregnancy, and the presence of abundant abdominal fat or gas in the intestine, may lead to a more difficult diagnosis. In these cases, abdomen and pelvis MRI, whose sensitivity and specificity are very high in the diagnosis of acute appendicitis (100% and 94–100% respectively), may be crucial for the diagnosis [72,73,74,75].

#### 3.9.3. Pancreatic and Biliary Pathology

During pregnancy there is an increased incidence rate of cholelithiasis (2–4% of pregnant women), linked to hormonal variations in pregnancy, which may result in a reduced mobility of the gallbladder and increased saturation of cholesterol in the bile. The most common presenting symptoms are nausea; vomiting; anorexia; right upper quadrant pain; and increased liver enzymes, alkaline phosphatase, and gamma-glutamil transferase (GGT). Additionally, in this case, the US represents the first-choice diagnostic method for pancreatic biliary pathology as it is endowed with high sensitivity and, above all, specificity in the diagnosis of acute cholecystitis (respectively 65% and 89%). However, MRI allows one to obtain a better visualization of the bile ducts and of any complications deriving from acute cholecystitis, such as perforation, pericholecystic abscess formation, and ascending cholangitis. Moreover, MRI also offers a better pancreas study, which is important to exclude possible pancreatitis, in some cases of not simple visualization to the US due to the presence of intestinal gas, and any complications, such as necrosis, pancreatic pseudocysts, and splenic vein thrombosis [76,77].

#### 3.9.4. Urolithiasis

Urolithiasis is one of the most common causes of abdominal pelvic pain in pregnancy, especially during the second and third trimesters. Symptoms include flank pain, nausea, vomiting, fever, and haematuria. A delayed diagnosis may result in complications, including pyelonephritis, reported in 0.5% of all pregnancies. The US is the first-choice diagnostic test as it is highly sensitive. However, in cases of equivocal results, MRI may be required [78,79].

#### 3.9.5. Neurological Conditions

There are numerous indications for performing an MRI during pregnancy for neurological pathologies; among these, the most common are acute headache, pregnancy-related spinal problems (i.e., pregnancy-related LBP, osteoporotic compression fractures, and symptomatic vertebral hemangioma), spinal cord injury, and brain tumors.

Acute headache is a very common neurological disorder during pregnancy, affecting about one in three pregnant women [80,81]. The differential diagnosis is between a benign headache and secondary headaches, which can endanger the life of the pregnant woman and the fetus, and includes hypertensive disorders, intracranial haemorrhage, stroke, cerebral vein thrombosis (CVT), or infections. Prompt diagnosis and possible treatment are therefore essential [80]. In recent years, the use of radiological imaging in pregnancy, in case of suspected secondary headache, which mainly makes use of MRI and provides contiguous orthogonal slices of the whole brain and excellent spatial and contrast resolution even without the use of contrast agents, has increased exponentially [3,82,83,84].

LBP is one of the most common and disabling problems during pregnancy, with a prevalence exceeding 50% [85,86]. A differential diagnosis is made when any other pathologies are present, such as osteoporotic or tumoral compression fractures, or Lumbar disc herniation, which may require prompt treatment, in order to avoid permanent neurological deficits [87].

Brain tumors are rare during pregnancy, the annual incidence is the same as for women of reproductive age (2.0 to 3.2 new cases per 100,000 people). The most frequent symptoms are headache; nausea and/or vomiting; and focal neurological deficits such as seizures, hemiparesis, and visual changes. Being very nonspecific during pregnancy, they risk leading to a late diagnosis. In light of this, any neurological deficit during worsening pregnancy should be suspected, without an otherwise known cause, and eventually imaging methods such as MRI should be resorted to [88,89].

MRI is also fundamental in the management and monitoring of multiple sclerosis (MS) during pregnancy. Indeed, during pregnancy, MS patients should be followed for any signs of illness aggravation [90]. If disease reactivation is suspected during pregnancy, it is safe to use low-field-strength MRI (1.5 Tesla) without contrast [90,91]. MRI, on the other hand, should only be considered if it is absolutely necessary, and the results may have therapeutic implications. All other MS diagnostic procedures, such as neurophysiological tests and lumbar puncture, are safe to carry out during pregnancy but should only be performed if they are absolutely necessary for the diagnosis [90].

#### 3.9.6. Cancer

The incidence of cancer during pregnancy is estimated at 1/1000 pregnancies. The most commonly diagnosed tumors during pregnancy are breast cancer, haematological cancers, cervical cancers, and melanoma. A cohort study analyzed 1170 women with pregnancy-associated cancers, of which 67% started treatment as early as pregnancy. Among these, the most common cancer was breast cancer (39%), followed by gynaecological ones (20%) [92,93,94].

Early diagnosis remains a major issue during pregnancy as well as during other phases of the life. Indeed, in case of initial and subclinical lesions, it is not possible to reach a diagnosis without screening campaign, which may increase the rate of early diagnosis and consequently improve prognosis. With regard to clinical evaluable, and self-palpable breast masses, US appears to be the imaging exam of first choice, showing very high sensitivity for pregnancy-associated breast carcinoma. Other complementary methods, useful in case of palpable breast mass, are mammography and unenhanced MRI [95]. In particular, the latter, not using ionizing radiation and having an excellent contrast resolution, is particularly useful for the diagnosis of breast cancer. The use of whole-body diffusion-weighted MRI for oncological staging of pregnant patients has also recently been proposed, demonstrating results that can be combined with GBCAs at the same MRI, although not using intravenous contrast [96,97,98,99].

US still appears to be the first-choice imaging exam in case of suspicion of gynecological tumors but is not always useful for diagnosis due to the profound anatomical changes during pregnancy. The presence of the pregnant uterus and the fetus can mask the pelvic organs from the ultrasound beam, making the possible diagnosis and staging of the tumor during pregnancy very complex. Here, too, MRI can be very useful as it provides multiplanar imaging and excellent soft tissue contrast at the pelvis [100].

#### 3.9.7. Other Non-Obstetric Causes

Other less common non-obstetric causes of acute abdominal pain in which MRI may increase diagnostic sensitivity include bowel obstruction, inflammatory bowel disease, HELLP (Hemolysis, Elevated Liver enzymes and Low Platelets) syndrome, and neoplasia [101,102].

### 3.10. Main Fetal Indications for MRI during Pregnancy

The fetal indications for performing MRI are less frequent than maternal; among these most are found in central nervous system (CNS), face and neck, and chest and abdomen (Table 2) [103]. Although US is the method of choice for fetal screening, MRI can add significantly to the diagnosis and management of congenital abnormalities. Indeed, when fetal anomalies are detected by US, the MRI can efficiently either confirm or reject the results, demonstrating its high value for both prenatal diagnosis and perinatal, as well as for management.

#### 3.10.1. CNS Anomalies

US is very useful in prenatal diagnosis of developmental and acquired intracranial anomalies; however, MRI shows a panoramic view of the whole brain and subarachnoid space without limits given by the skull and is crucial for any suspicions detected by the US or to evaluate any pathologies, which are difficult to appreciate ultrasonographically [103,104,105].

#### 3.10.2. Face and Neck

US is the imaging method of first choice to detect any abnormalities of the head and neck, such as cleft lip and palate, micrognathia or retrognathia, craniosynostosis, cephaloceles, vascular anomalies, tumors, microphthalmia, thyroid anomalies, or oropharyngeal and neck masses. In some cases, however, when the fetus is in positions where the head and neck are not assessable by ultrasound, as rotated or covered by the limbs, MRI can be very useful [106,107,108].

#### 3.10.3. Chest

In this case, US is the main screening method for thoracic abnormalities, such as diaphragmatic hernia, cystic adenomatoid malformation, bronchopulmonary seizure, or the presence of other cysts or masses that can lead to pulmonary hypoplasia and fetal death if not identified early [109,110,111,112,113]. Moreover, US can identify also cardiac anomalies or malformations [114,115,116]. In case of diagnostic doubt, MRI can be of considerable help.

#### 3.10.4. Abdomen

US and MRI successfully demonstrate, usually after 18 weeks, abdominal anomalies, such as oesophageal and bowel atresia, intra-abdominal masses, abdominal wall defects, bowel obstruction, bowel perforation and meconium peritonitis, renal agenesis or ectopy, duplication of the collecting systems, urinary tract dilatation ureteroceles, severe vesical-ureteral reflux, megaureter, bladder outlet obstruction, or cloacal anomalies [117,118,119,120].

#### 3.10.5. Limbs

Any limb anomalies, such as abnormal finger position, agenesis, syndactyly, polydactyly, or phocomelia, can be easily identified through the use of US and MRI [121].

### 3.11. GBCAs during Lactation

GBCAs administered intravenously during pregnancy are excreted in breast milk over the next 24 h in very small amounts, less than 0.04% of the initial administered dose, due to the reduced binding to milk proteins. In addition, less than 1% of this percentage will be absorbed from the child’s gastrointestinal tract, much less than the recommended doses for use in pediatric patients [122,123].

Until a few years ago, a suspension of breastfeeding for about 24 h after administration of GBCAs was recommended as a precaution. Currently, however, no studies have highlighted any adverse effects for the child after administering GBCAs to the mother. Therefore, in light of the basic final absorbed dose and safety for the baby, the ACR and the ACOG recommend not discontinuing breastfeeding as it may lead to early weaning [124]. Indeed, whenever possible, it is preferable to postpone contrast-enhanced MRI at the end of pregnancy, performing it even during breastfeeding.

### 3.12. Liposomal Gadolinium Nanoparticle Contrast Agents

In a recent published preclinical study, in animal models, Shetty et al. showed that a long circulating liposomal nanoparticle-based blood-pool gadolinium contrast agent (liposomal-Gd) (Figure 1) does not penetrate the placental barrier, due to its size, and therefore does not expose the fetus to the contrast agent during pregnancy, preserving it from any hypothetical risks [125]. In fact, liposomes cross the placental barrier in a very limited way, due to their larger size (diameter 100–150 nm, about three orders of magnitude larger than free molecules) and the presence of a Polyethylene glycol (PEG) coating on the surface of the liposomes. Indeed, these characteristics prevent the binding to the walls of the vascular endothelium and therefore the active trans-placental transport. Regarding its pharmacokinetics, liposomal-Gd has a half-life of approximately 18–24 h and is then cleared from the blood pool by the reticuloendothelial system of the liver and spleen [126,127].

Liposomal-Gd was more effective than conventional macrocyclic-based GBCAs for the visualization of placental margins and retro-placental clear space during the second half of gestation in rodent animal models [128,129,130,131].

## 4. Conclusions

MRI is a useful tool for evaluating numerous obstetric and non-obstetric conditions during pregnancy as it has excellent spatial and contrast resolution, free of ionizing radiation, and non-operator-dependent results. However, its use during pregnancy raises a number of medical, ethical, and legal assessments. To date, in fact, there are no human studies that have highlighted any negative effects on the fetus following exposure to MRI without contrast agent during any trimester of pregnancy; however, regarding the use of contrast agent during this period, as also suggested by international guidelines, there is a tendency to adopt a more cautious attitude. Indeed, based on the data in the literature, the consensus is that GBCAs are linked to a little or no risk for the mother, but it is less conclusive on the safety profile for the fetus. In fact, there is only one large cohort study showing a slight increased risk of neonatal death associated with the use of GBCAs during pregnancy. More studies are needed to confirm these data and to better clarify the role of GBCAs during pregnancy. Therefore, since there is no clear scientific evidence that the contrast agent has no adverse effects on the fetus, its use in the event of an ascertained, or even suspected, pregnancy must be carefully evaluated by the radiologist and the requesting physician, approved only if it can affect the therapeutic management of the mother and/or the fetus, and it cannot be postponed to the end of the pregnancy or after having explained in a clear and detailed way to the patient any risks and benefits deriving from its use, and finally by having acquired informed consent by the patient.

Unenhanced MRI avoids the risk of short-term risks, observed with the use of GBCAs, such as allergic reactions and non-allergic reactions (nausea and vomiting); of severe reactions to the contrast agents, which are characteristic of pregnancy (recurrent late decelerations, prolonged fetal bradycardia on fetal heart tracing, and preterm labor); and of long-term risks, such as NSF and retained gadolinium but, especially in the oncology field, which does not allow one to appreciate the presence of central or peripheral nodular enhancement in a mass of unknown etiology, which can be very useful to help distinguish between a possible tumor, of a benign or malignant nature, and any simple cysts or fluid collections [132]. The use of MRI with GBCAs, in addition to the correct identification of the lesion, can help in the choice of the most adequate treatment and prognostic indication. In the brain, contrast-enhanced MRI is considered the “gold standard” as it greatly improves the detection of primary tumors and metastases, including metastatic leptomeningeal disease in the brain and spinal canal [133]. In recent years, new GBCAs (such as liposomial-Gd) have been developed, showing a promising safety profile in pregnancy. Indeed, these new contrast agents, which do not cross the placental barrier and do not expose the fetus to any risks, may represent a better choice during pregnancy, as already shown in animal models.

Hence, more studies are needed to better clarify the potential key role of these new contrast agents; their safety has also been evaluated in studies on human models, and to confirm their safety during pregnancy. Moreover, we believe that official and widely approved guidelines should be implemented regarding the use of unenhanced and contrast-enhanced MRI during pregnancy, better describing all cases in which it should be used and when it should be avoided in favor of other diagnostic imaging examinations.

## Figures and Tables

**Figure 1 jpm-12-00009-f001:**
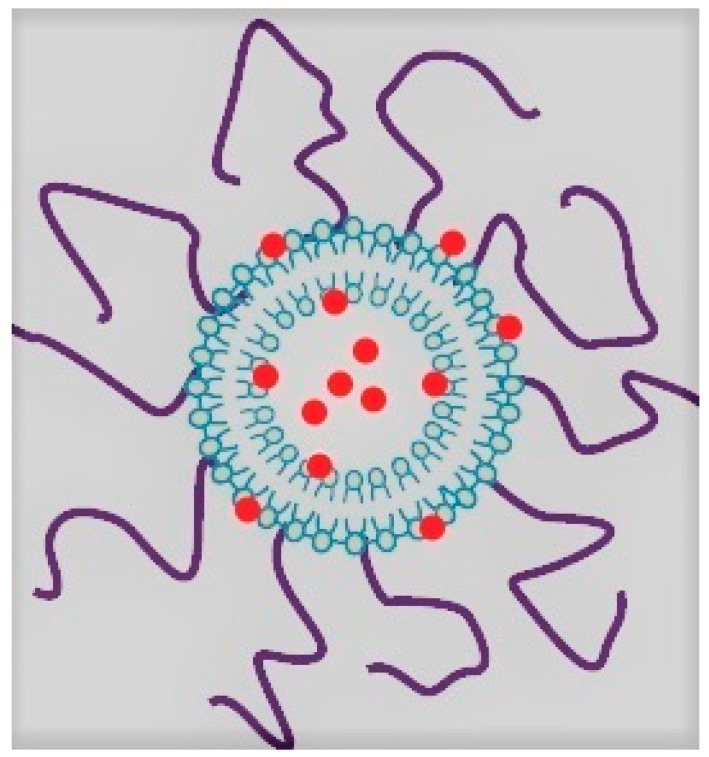
Liposomal nanoparticle-based blood-pool gadolinium contrast agent structure. Low molecular-weight gadolinium-chelated (in red) is inserted, both core-encapsulated and surface-conjugated, in liposome structures made of phospholipid bilayer (in light blue). The liposomes are also coated with PEG (in purple) and thus avoid penetration through placental barrier.

**Table 1 jpm-12-00009-t001:** Commercially available gadolinium-based contrast agents (GBCAs) in Europe approved during pregnancy.

Trade Name	Marketing Authorisation Holder	Compound	Chemical Structure	Use
Dotarem^®^	Guerbet Diagnostic Imaging	Gadoterate meglumine	Macrocyclic	Intraarticular/Intravenous
Gadovist^®^	Bayer Pharmaceuticals	Gadobutrolo	Macrocyclic	Intravenous
Magnevist^®^	Bayer Pharmaceuticals	Gadopentetate dimeglumine	Linear	Intraarticular
Multihance^®^	Bracco Imaging	Gadobenate dimeglumine	Linear	Intravenous
Primovist^®^	Bayer Pharmaceuticals	Gadoxetate disodium	Linear	Intravenous
Prohance^®^	Bracco Imaging	Gadoteridol	Macrocyclic	Intravenous

**Table 2 jpm-12-00009-t002:** Fetal main indications for emergency MRI during pregnancy.

Site	Indications
CNS anomalies	Ventriculomegaly, hemorrhages, lissencephaly, polymicrogyria/pachygyria, gray matter heterotopias, cortical dysplasias, and neural tube defects (e.g., spina bifida/diastematomyelia).
Face and palate	In cases in which there is a significant risk of associated brain abnormalities.
Neck masses	Neck masses could impair the airway leading to asphyxia at birth.
Chest	Congenital diaphragmatic hernias.
Abdomen	Abdominal masses or bowel pathologies, including obstruction and atresia.

CNS: central nervous system.

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
