# Peer review of "MRI in Pregnancy and Precision Medicine: A Review from Literature"

_jpm, 2021, doi:10.3390/jpm12010009_

Round 1
Reviewer 1 Report
This work is well organized and contain useful information about MRI safety during pregnancy.
1) In this document is mostly focused on the use of contrast agents during imaging. I would like to suggest to mention in the document about the specific absorption rate (SAR), as it is also an important part for safety in MRI.
2) a minor comment would be to make the first word of the tables capital letters.
3) the references (18-20) are mentioned two times in the same paragraph. It would be better to place the reference number at the sentence that is more related to.
Author Response
1) In this document is mostly focused on the use of contrast agents during imaging. I would like to suggest to mention in the document about the specific absorption rate (SAR), as it is also an important part for safety in MRI.
R: First of all I thank the Reviewer for the valuable comments.
I have added some considerations on SAR to the paper.
2) a minor comment would be to make the first word of the tables capital letters.
R: I have made the first word of the tables capital letters.
3) the references (18-20) are mentioned two times in the same paragraph. It would be better to place the reference number at the sentence that is more related to.
R: I have removed the repetition of the reference number.
Reviewer 2 Report
This article aims in performing literature review focusing on contrast and non-contrast MRI use during pregnancy.
Overall the topic is of interest and importance – especially when focusing on Personalized Medicine. While the authors have provided a wide review, several topics are missing from the text and should be added to the article before publications.
Major comments:
- Major issue: An additional mentioning to the utilization of MRI during lactation which usually accompanies articles related to pregnancy would add depth and interest to the paper.
- There is no referring to maternal neurological indications for emergency MRI (except for abstract). This whole section should be added.
- There is no referring to fetal indications for MRI. This whole section should be added.
- Another section for non-emergent body indications such as breast cancer MRI during pregnancy is missing.
- It would strengthen the paper if the authors would discuss the physiological changes that comes with pregnancy and their reflection on MRI.
- Unenhanced MRI should be discussed more thoroughly in terms of pros and cons vs contrast MRI, when suggested as an alternative during pregnancy. When is it satisfactory and when it is inferior to contrast enhanced MRI during pregnancy?
- The conclusions section is missing clear outlook.
- Missing ref are suggested.
Body
3/1 Non Contrast MRI during Pregnancy:
The statement that “no studies showing that MRI causes fetal harm” is too strong in the view of the later statement that “…during pregnancy was associated with reduced birth weight and increased stillbirth”.
Please revise.
Also ref 19, should receive more attention, as it highlights the differences in MRI with and without Gad on fetal outcome, including increased still birth risk only when gad is injected.
Also another paragraph dedicated to the use of MRI in the first trimester is warranted.
3/2 Gadolinium-based Contrast Agents (GBCAs) MRI during Pregnancy
The following important ref. is missing:
Bird et al. First-Trimester Exposure to Gadolinium-based Contrast Agents: A Utilization Study of 4.6 Million U.S. Pregnancies. Radiology. 2019.
3.8. Obstetric Causes
Well written
3.9. Non Obstetric Causes
- This section is very thin and missing lots of data. There is no referring to any neurological indication for MRI of both the central and peripheral nervous system. Although rare MRI may be utilized in patients with acute neurological symptoms, exacerbation of known neurological disease during pregnancy, neuro-oncological evaluation, trauma evaluation to spinal cord, and more…
To name a few missing ref:
Bianca Raffaelli. Brain imaging in pregnant women with acute headache. J Neurol. 2018 Aug;265(8):1836-1843.
Bonfield CM. Pregnancy and brain tumors. Neurol Clin. 2012
Han IH. Pregnancy and spinal problems. Curr Opin Obstet Gynecol. 2010
- The role of fetal MRI is somehow mentioned in the abstract but is not reviewed at all. Clearly there is a strong evidence supporting the role of MRI in evaluation of fetal development for both fetal brain and non brain indications.
Pugash D et al. Prenatal ultrasound and fetal MRI: The comparative value of each modality in prenatal diagnosis. EJR 2008
Cassart M Fetal Body Imaging: When is MRI Indicated?
J Belg Soc Radiol. . 2017
- There are also missing referring to maternal non neurological indications for MRI. For example diagnosis and staging of breast cancer using MRI:
Nissan N, et al. Noncontrast Breast MRI During Pregnancy Using Diffusion Tensor Imaging: A Feasibility Study. J Magn Reson Imaging. 2019
Nissan N, Breast MRI Without Contrast Is Feasible and Appropriate During Pregnancy.J Am Coll Radiol. 2019
Han SN, Feasibility of whole-body diffusion-weighted MRI for detection of primary tumour, nodal and distant metastases in women with cancer during pregnancy: a pilot study. Eur Radiol. 2018
Peccatori FA, Whole body MRI for systemic staging of breast cancer in pregnant women.Breast. 2017
- 11. Liposomal Gadolinium Nanoparticle Contrast Agents
Besides the mentioning that nanoparticle-based Gadolinium contrast agent does not penetrate the placental barrier, the whole safety profile of this contrast agent should be discussed. Is it toxic to the mother? Is it ready to clinical use?
Also Figure 1 caption is very limited. The authors should specifically state what microstructure is presented in the figure and its relevance to the contrast agent distribution profile.
- Conclusions
This section lacks a clear outlook by the research group. The conclusions are too general and vague. Here a more precise summary should have been provided. what is the authors view of future MRI indications and utilization? (besides nano particles that are far from clinical utilization).
In addition, while discussing the concerns about Gad use in pregnancy, advantages in unenhanced methods should be discussed and their current limitations as compared with contrast MRI.
Finally, the physiological changes related to pregnancy and their influence on MRI were not discussed at al. This topic is of interest and goes beyond safety issues.
Author Response
Reviewer 2
“This article aims in performing literature review focusing on contrast and non-contrast MRI use during pregnancy.
Overall the topic is of interest and importance – especially when focusing on Personalized Medicine. While the authors have provided a wide review, several topics are missing from the text and should be added to the article before publications.”
Answer: Many thanks for Your comments and appreciation. We followed all Your suggestions as follow
Major comments:
- “Major issue: An additional mentioning to the utilization of MRI during lactationwhich usually accompanies articles related to pregnancy would add depth and interest to the paper.”
Answer: A focus on MRI during lactation has been added ( line 367-377)
- “There is no referring to maternal neurological indications for emergency MRI (except for abstract). This whole section should be added.”
Answer: A focus on MRI during lactation has been added ( line 284-326)
- “There is no referring to fetal indications for MRI. This whole section should be added.”
Answer: A focus on fetal MRI has been added (line 330-353)
- “Another section for non-emergent body indications such as breast cancer MRI during pregnancy is missing.”
Answer: A focus on this topic has been added (line 308-326)
- “It would strengthen the paper if the authors would discuss the physiological changes that comes with pregnancy and their reflection on MRI.”
Answer: A focus on this topic has been added (line 78-88)
- “Unenhanced MRI should be discussed more thoroughly in terms of pros and cons vs contrast MRI, when suggested as an alternative during pregnancy. When is it satisfactory and when it is inferior to contrast enhanced MRI during pregnancy?”
Answer: A discussion on this topic has been added (line 455-466)
“The conclusions section is missing clear outlook.”
Answer: The conclusion section has been improved as suggested.
- “Missing ref are suggested.”
Answer: Missing references have been added as suggested
“Body
3/1 Non Contrast MRI during Pregnancy:
“The statement that “no studies showing that MRI causes fetal harm” is too strong in the view of the later statement that “…during pregnancy was associated with reduced birth weight and increased stillbirth”.
Please revise.”
Answer: The sentence has been changed as suggested.
-“Also ref 19, should receive more attention, as it highlights the differences in MRI with and without Gad on fetal outcome, including increased still birth risk only when gad is injected.”
Answer: The text was implemented an suggested
Also another paragraph dedicated to the use of MRI in the first trimester is warranted.
Answer: A discussion on this topic has been added (line 119-122)
3/2 Gadolinium-based Contrast Agents (GBCAs) MRI during Pregnancy
The following important ref. is missing:
Bird et al. First-Trimester Exposure to Gadolinium-based Contrast Agents: A Utilization Study of 4.6 Million U.S. Pregnancies. Radiology. 2019.
Answer: The cited reference was added
3.8. Obstetric Causes
Well written
3.9. Non Obstetric Causes
- This section is very thin and missing lots of data. There is no referring to any neurological indication for MRI of both the central and peripheral nervous system. Although rare MRI may be utilized in patients with acute neurological symptoms, exacerbation of known neurological disease during pregnancy, neuro-oncological evaluation, trauma evaluation to spinal cord, and more…
Answer: A discussion on the cited topics has been added
To name a few missing ref:
Bianca Raffaelli. Brain imaging in pregnant women with acute headache. J Neurol. 2018 Aug;265(8):1836-1843.
Bonfield CM. Pregnancy and brain tumors. Neurol Clin. 2012
Han IH. Pregnancy and spinal problems. Curr Opin Obstet Gynecol. 2010
Answer: Missing references were added
- The role of fetal MRI is somehow mentioned in the abstract but is not reviewed at all. Clearly there is a strong evidence supporting the role of MRI in evaluation of fetal development for both fetal brain and non brain indications.
Pugash D et al. Prenatal ultrasound and fetal MRI: The comparative value of each modality in prenatal diagnosis. EJR 2008
Cassart M Fetal Body Imaging: When is MRI Indicated?
J Belg Soc Radiol. . 2017
Answer: A discussion on this topic, and Missing references has been added.
- There are also missing referring to maternal non neurological indications for MRI. For example diagnosis and staging of breast cancer using MRI:
Nissan N, et al. Noncontrast Breast MRI During Pregnancy Using Diffusion Tensor Imaging: A Feasibility Study. J Magn Reson Imaging. 2019
Nissan N, Breast MRI Without Contrast Is Feasible and Appropriate During Pregnancy.J Am Coll Radiol. 2019
Han SN, Feasibility of whole-body diffusion-weighted MRI for detection of primary tumour, nodal and distant metastases in women with cancer during pregnancy: a pilot study. Eur Radiol. 2018
Peccatori FA, Whole body MRI for systemic staging of breast cancer in pregnant women.Breast. 2017
Answer: A discussion on this topic, and Missing references has been added.
Conclusions
This section lacks a clear outlook by the research group. The conclusions are too general and vague. Here a more precise summary should have been provided. what is the authors view of future MRI indications and utilization? (besides nano particles that are far from clinical utilization).
In addition, while discussing the concerns about Gad use in pregnancy, advantages in unenhanced methods should be discussed and their current limitations as compared with contrast MRI.
Answer: Conclusions were better summarized and discussed as suggested
Finally, the physiological changes related to pregnancy and their influence on MRI were not discussed at al. This topic is of interest and goes beyond safety issues.
Answer: A discussion on this topic, has been added.
Round 2
Reviewer 2 Report
I would like to commend the authors for their thorough and wide scale review, in a topic of great interest. The additions of the revised paper have strengthen the manuscript. Few more minor changes are requested before the paper is ready for publication.
- Introduction
“Moreover, during pregnancy there are various physiological changes…” – good addition, though reference(s) is missing.
3.Body
Non contrast enhanced – excellent.
3.9.5 Neurological conditions –
Please add the following references as previously requested:
Bianca Raffaelli. Brain imaging in pregnant women with acute headache. J
Neurol. 2018 Aug;265(8):1836-1843.
Bonfield CM. Pregnancy and brain tumors. Neurol Clin. 2012
Han IH. Pregnancy and spinal problems. Curr Opin Obstet Gynecol. 2010
Also MRI is fundamental in the management and monitoring of multiple sclerosis (MS). Please refer to disease specifically and discuss the policy of MS follow-up during pregnancy.
3.9.6. Cancer
“In the case of palpable breast mass, US appears to be the imaging exam of first choice, 335 showing a very high sensitivity (up to 100%) for pregnancy-associated breast carcinoma”.
- This statement, although accurate, is too strong and would make the reader believe that PABC is under control, while in reality is a dreadful disease, being diagnosed only until the lesion gets really large and easy for self palpable (and hence, 100% sensitivity..). The concept of early diagnosis is missed during pregnancy. Screening of breast cancer is a huge problem. Please revise the paragraph, although good, to deliver a more balance message.
While it appears you have made the required corrections and added references, the relevant ones are still missing. Actually references 85-90 are all unrelated to the text (pyelonephritis, frequency of cancer, survival, fetal brain, fetal CNS disorder – NONE is related to unehanced/diffusion/whole body MRI in pregnancy).
Please check on 1 by 1 basis that all references are in place.
Then, please add the following references, as previously requested.
Nissan N, et al. Noncontrast Breast MRI During Pregnancy Using Diffusion Tensor Imaging: A Feasibility Study. J Magn Reson Imaging. 2019
Nissan N, Breast MRI Without Contrast Is Feasible and Appropriate During Pregnancy. J Am Coll Radiol. 2019
Han SN, Feasibility of whole-body diffusion-weighted MRI for detection of primary tumour, nodal and distant metastases in women with cancer during pregnancy: a pilot study. Eur Radiol. 2018
Peccatori FA, Whole body MRI for systemic staging of breast cancer in pregnant women.Breast. 2017
3.10 Fetal indications
Is it really “emergency MRI”? I wouldn’t say so. Although important, it is always electively performed. In this paragraph, a more “zoom-out” perspective is needed, with some discussion on the additive value of fetal MRI. What diagnostic edge it provides that gynecological US is inferior to? For which indications, or diagnosis it is specifically instrumental.
3.11. GBCAs During Lactation
Good addition. Here it might be worth discussing that in some indications it is advised to postpone MRI from pregnancy to post-partum.
3.12. Liposomal Gadolinium Nanoparticle Contrast Agents
Figure caption is unsatisfactory. Since none of us is chemist, and you chose to display this figure, please make the effort to provide accessibility to the topic, related to the microstructure of “liposome”. Please elaborate briefly what are the sub units in the different colors/shapes and how it is related to the Gad pharmacokinetics.
Author Response
First of all, I would like to thank’s the reviewer for the important support and suggestions given to improve our manuscript.
Your contribution was fundamental.
Herein we report a point-to-point letter describing all changes. Particularly:
“I would like to commend the authors for their thorough and wide scale review, in a topic of great interest. The additions of the revised paper have strengthen the manuscript. Few more minor changes are requested before the paper is ready for publication.”
1.Introduction
“Moreover, during pregnancy there are various physiological changes…” – good addition, though reference(s) is missing.
Answer: Many thanks for Your comment and suggestions. We have added missing references.
3.Body
“Non contrast enhanced – excellent.”
Answer: Thanks for Your comment
3.9.5 Neurological conditions –
“Please add the following references as previously requested:
Bianca Raffaelli. Brain imaging in pregnant women with acute headache. J
Neurol. 2018 Aug;265(8):1836-1843.
Bonfield CM. Pregnancy and brain tumors. Neurol Clin. 2012
Han IH. Pregnancy and spinal problems. Curr Opin Obstet Gynecol. 2010”
Answer: References has been added ( 88, 94,95)
“Also MRI is fundamental in the management and monitoring of multiple sclerosis (MS). Please refer to disease specifically and discuss the policy of MS follow-up during pregnancy.”
Answer: A focus on this topic has been added in the “Neurological conditions “ section ( line 315-322)
3.9.6. Cancer
“In the case of palpable breast mass, US appears to be the imaging exam of first choice, 335 showing a very high sensitivity (up to 100%) for pregnancy-associated breast carcinoma”.
- “This statement, although accurate, is too strong and would make the reader believe that PABC is under control, while in reality is a dreadful disease, being diagnosed only until the lesion gets really large and easy for self palpable (and hence, 100% sensitivity..). The concept of early diagnosis is missed during pregnancy. Screening of breast cancer is a huge problem. Please revise the paragraph, although good, to deliver a more balance message.”
Answer: This section has been modified in order to deliver a more balanced message.
“While it appears you have made the required corrections and added references, the relevant ones are still missing. Actually references 85-90 are all unrelated to the text (pyelonephritis, frequency of cancer, survival, fetal brain, fetal CNS disorder – NONE is related to unehanced/diffusion/whole body MRI in pregnancy).
Please check on 1 by 1 basis that all references are in place.
Then, please add the following references, as previously requested.
Nissan N, et al. Noncontrast Breast MRI During Pregnancy Using Diffusion Tensor Imaging: A Feasibility Study. J Magn Reson Imaging. 2019
Nissan N, Breast MRI Without Contrast Is Feasible and Appropriate During Pregnancy. J Am Coll Radiol. 2019
Han SN, Feasibility of whole-body diffusion-weighted MRI for detection of primary tumour, nodal and distant metastases in women with cancer during pregnancy: a pilot study. Eur Radiol. 2018
Peccatori FA, Whole body MRI for systemic staging of breast cancer in pregnant women.Breast. 2017”
Answer: All references have been checked 1 by 1 and have been placed in the correct order. Moreover, all suggested references have been added as required. (103,104,105,106)
3.10 Fetal indications
“Is it really “emergency MRI”? I wouldn’t say so. Although important, it is always electively performed. In this paragraph, a more “zoom-out” perspective is needed, with some discussion on the additive value of fetal MRI. What diagnostic edge it provides that gynecological US is inferior to? For which indications, or diagnosis it is specifically instrumental.”
Answer: The paragraph has been modified as suggested (line 350-354)
3.11. GBCAs During Lactation
Good addition. Here it might be worth discussing that in some indications it is advised to postpone MRI from pregnancy to post-partum.
Answer: We added this concept as suggested ( line 382-384)
3.12. Liposomal Gadolinium Nanoparticle Contrast Agents
Figure caption is unsatisfactory. Since none of us is chemist, and you chose to display this figure, please make the effort to provide accessibility to the topic, related to the microstructure of “liposome”. Please elaborate briefly what are the subunits in the different colors/shapes and how it is related to the Gad pharmacokinetics.
Answer: We better-elaborated figure caption as suggested. (line 826-830)